# A Review on Membrane Fouling Prediction Using Artificial Neural Networks (ANNs)

**DOI:** 10.3390/membranes13070685

**Published:** 2023-07-24

**Authors:** Waad H. Abuwatfa, Nour AlSawaftah, Naif Darwish, William G. Pitt, Ghaleb A. Husseini

**Affiliations:** 1Materials Science and Engineering Ph.D. Program, College of Arts and Sciences, American University of Sharjah, Sharjah P.O. Box 26666, United Arab Emirates; g00062257@alumni.aus.edu (W.H.A.); g00051790@alumni.aus.edu (N.A.); 2Department of Chemical and Biological Engineering, College of Engineering, American University of Sharjah, Sharjah P.O. Box 26666, United Arab Emirates; ndarwish@aus.edu; 3Chemical Engineering Department, Brigham Young University, Provo, UT 84602, USA; pitt@byu.edu

**Keywords:** artificial neural networks (ANNs), fouling, prediction, simulation, membranes

## Abstract

Membrane fouling is a major hurdle to effective pressure-driven membrane processes, such as microfiltration (MF), ultrafiltration (UF), nanofiltration (NF), and reverse osmosis (RO). Fouling refers to the accumulation of particles, organic and inorganic matter, and microbial cells on the membrane’s external and internal surface, which reduces the permeate flux and increases the needed transmembrane pressure. Various factors affect membrane fouling, including feed water quality, membrane characteristics, operating conditions, and cleaning protocols. Several models have been developed to predict membrane fouling in pressure-driven processes. These models can be divided into traditional empirical, mechanistic, and artificial intelligence (AI)-based models. Artificial neural networks (ANNs) are powerful tools for nonlinear mapping and prediction, and they can capture complex relationships between input and output variables. In membrane fouling prediction, ANNs can be trained using historical data to predict the fouling rate or other fouling-related parameters based on the process parameters. This review addresses the pertinent literature about using ANNs for membrane fouling prediction. Specifically, complementing other existing reviews that focus on mathematical models or broad AI-based simulations, the present review focuses on the use of AI-based fouling prediction models, namely, artificial neural networks (ANNs) and their derivatives, to provide deeper insights into the strengths, weaknesses, potential, and areas of improvement associated with such models for membrane fouling prediction.

## 1. Introduction

Membrane-based processes have emerged as a promising and efficient approach for water treatment due to their ability to remove a wide range of contaminants, including suspended solids, dissolved organic matter, and inorganic salts [1]. Membrane-based processes utilize semi-permeable membranes capable of the selective separation of different constituents in liquid or gaseous feed streams. They have several advantages over conventional water treatment methods, including lower energy consumption, a smaller footprint, and higher removal efficiency [1]. Pressure-driven membrane processes are widely used technologies for water treatment, desalination, and wastewater reuse [2]. These processes involve using a semi-permeable membrane that separates contaminants from water based on size, charge, or hydrophobicity. Pressure-driven membrane processes, which usually employ porous membranes, include four important and widely used membrane processes, which are microfiltration (MF), ultrafiltration (UF), nanofiltration (NF), and reverse osmosis (RO) [2]. These four processes, in the order mentioned, are defined by their decreasing pore size. The membrane in these processes acts only as a barrier between a feed stream and a permeate stream.

MF membranes usually come with a pore size of 0.1–10 µm and are used to remove suspended solids, bacteria, and some viruses [3]. UF processes use membranes with a pore size of 0.001–0.1 µm and are used for the removal of macromolecules such as proteins, polysaccharides, and viruses. UF is commonly used as a pretreatment step in water treatment systems, particularly in surface water or wastewater treatment. It can also be used as a standalone treatment process to produce high-quality drinking water or for the removal of specific contaminants, such as endotoxins [4,5] or macromolecules [6,7]. It is an effective and energy-efficient water treatment process that can produce high-quality water with low consumption of chemicals and minimal waste generation [8]. NF membranes have a pore size of 0.001–0.01 µm and are used to remove divalent ions, such as calcium and magnesium. RO membranes have a pore size of less than 0.001 µm and are used to exclude monovalent ions, such as sodium and chloride, as well as organic molecules and bacteria [9].

Membrane fouling is a major challenge in these processes, which can impair the performance of the membrane and increase operational costs [10]. Therefore, it is important to understand, diagnose, predict, and mitigate fouling occurrences for the sustainable and efficient operation of membrane-based processes [11]. Researchers have heavily relied on mathematical models that utilize membrane flux, transmembrane pressure, pollutant rejection, and other operating parameters to better understand and forecast membrane fouling behavior in water and wastewater treatment processes [12]. While such traditional models are well established, they have several drawbacks that limit their usefulness in predicting fouling in different membrane systems and operating conditions (e.g., limited flexibility, data-intensive, use of simplified assumptions, etc.) [12]. This highlights the need to develop more effective approaches to better predict the membrane-based processes’ performance and fouling propensity. Artificial intelligence (AI)-based models have shown promising results in water membrane research and are expected to play an increasingly important role in membrane system design, process optimization, and fouling prediction [13,14,15,16]. AI-based models can learn complex patterns and relationships in large datasets to improve the accuracy of predictions and enable the more efficient optimization of membrane systems, which may not be possible using conventional models [16,17].

Unlike other existing reviews that focus on mathematical models [18,19,20] or broad AI-based simulations [17], the present review focuses on the use of AI-based fouling prediction models, namely, artificial neural networks (ANNs) and their derivatives, to provide deeper insights into the strengths, weaknesses, potential, and areas of improvement associated with such models for membrane fouling prediction. This paper provides an overview of the different types of membrane fouling and mitigation strategies, a brief review of existing mathematical models, a basic introduction to ANNs, and a thorough review of the pertinent literature.

## 2. Fouling Types and Mitigation Strategies

Membrane fouling is defined as the accumulation of unwanted (organic, inorganic, and particulate) materials on the surface or within the pores of a membrane, which leads to a decline in its flux performance over time [12]. The fouling layer can be either reversible or irreversible, depending on the nature of the foulants and the severity of the fouling. Generally, fouling formation can be either physical or chemical. Physical fouling occurs when particles or fibers larger than the pore size of the membrane accumulate on its surface, leading to a reduction in its permeability. The mechanism of physical fouling involves the deposition of particles on the membrane surface due to gravitational settling, inertial impaction, and diffusion. It can also occur due to concentration polarization, which is the buildup of solutes near the membrane surface due to the concentration gradient across the membrane, leading to the precipitation of salts or other materials near the membrane surface and further contributing to fouling. Effective pretreatment strategies are necessary to mitigate physical fouling in membrane processes, such as sedimentation, filtration, and coagulation [21].

On the other hand, chemical fouling can be caused by the reaction of chemical species in the feed stream with the membrane surface, forming an insoluble layer that obstructs the membrane pores. The mechanism involves the adsorption of ions on the membrane surface, followed by the nucleation and growth of the insoluble layer. It can be classified into scaling and precipitation fouling [12]. Scaling fouling occurs when dissolved salts or minerals in the feed stream precipitate on the membrane surface or within its pores, forming a scale layer [22]. Precipitation fouling occurs when dissolved compounds in the feed stream react with each other or the membrane material, forming an insoluble precipitate that obstructs the membrane pores [22]. Several factors promote chemical fouling, such as high pH, temperature, ionic strength, and divalent cations (e.g., calcium and magnesium) [12]. Inorganic salts, such as calcium carbonate and calcium sulfate, are common foulants in many membrane-based separation processes. For instance, calcium ions tend to react with carbonate and bicarbonate ions present in the water, leading to the precipitation of calcium carbonate (CaCO_3_) or calcium bicarbonate (Ca(HCO_3_)_2_) [23]. These sediments can adhere to the membrane surface and form a scale layer, reducing the membrane’s permeability and causing flux decline. Calcium scaling is widespread in water sources with high calcium hardness, such as groundwater or water supplies containing significant amounts of dissolved calcium. Likewise, poly aluminum chloride (PAC) is a commonly used coagulant in water and wastewater treatment processes [24]. It is added to remove suspended particles, colloids, and organic matter by forming flocs that can be easily separated. However, if PAC is not effectively removed or controlled, it can contribute to fouling in membrane systems. PAC contains aluminum hydroxide compounds, which can form gel-like particles or precipitates that can adsorb onto the membrane surface, accumulate, and form a fouling layer [24]. Other common foulants include silica, iron, and aluminum compounds [22,25]. Effective pretreatment strategies are necessary to mitigate chemical fouling, such as pH adjustment, antiscalant dosing, and membrane surface modification. Regular membrane surface cleaning can also help remove accumulated foulants and prolong the membrane’s lifespan. Therefore, monitoring the extent and nature of chemical fouling is essential to determine the appropriate cleaning and maintenance protocols [21,25].

Furthermore, fouling can be classified into different types based on the nature of the foulants, which include inorganic fouling, particulate fouling, organic fouling, and biofouling [11,18]. Inorganic fouling occurs when inorganic compounds, such as salts, metals, and silica, precipitate or crystallize on the membrane surface or within its pores. Inorganic compounds can come from various sources, such as seawater, groundwater, and industrial effluents. The mechanism of inorganic fouling is mainly attributed to the thermodynamic and kinetic properties of the inorganic compounds, which can be influenced by the feedwater chemistry and operating conditions [25]. The formation of inorganic deposits can be triggered by temperature, pressure, pH, and concentration changes, which can induce the supersaturation and nucleation of the compounds. Once the deposits are formed, they can grow and agglomerate, forming a cake layer on the membrane surface or within its pores [25]. While chemical fouling can provide a surface for inorganic fouling, it is not always necessary for inorganic fouling to occur. Inorganic fouling can also occur on clean surfaces, especially in environments where the materials have a suitable substrate and environmental conditions for deposition [11,25,26].

Particulate fouling occurs when suspended or colloidal particles accumulate on the membrane surface or within its pores [27]. The particles can have various sizes and shapes, and their accumulation can reduce the membrane permeability and increase its pressure drop. The primary causes of particulate fouling are feedwater quality, operating conditions, and membrane properties. The mechanisms of particulate fouling are complex and depend on several factors, including particle size, shape, surface chemistry, and hydrodynamic conditions. The particles can deposit on the membrane surface due to electrostatic and van der Waals forces and their attachment can be enhanced by hydrodynamic shear forces [27]. Once the particles are attached to the membrane surface, they can form a cake layer that further traps other particles and reduces the membrane’s permeability [27].

Furthermore, organic fouling occurs when organic molecules, such as proteins, polysaccharides, and humic substances, adsorb onto the membrane surface and block its pores [28]. Organic molecules can be of natural or synthetic origin and can come from various sources, such as wastewater, food processing, and pharmaceuticals. The mechanism of organic fouling is mainly attributed to the adsorption of organic molecules onto the membrane surface, followed by their aggregation and the formation of a cake layer. The adsorption can be influenced both by the surface properties of the membrane—such as hydrophobicity, charge, and roughness—and by the properties of the organic molecules—such as size, shape, and charge density [28]. In addition, the presence of foulants can alter the membrane’s surface chemistry and induce surface-induced conformational changes in the organic molecules, which can further enhance their adsorption and aggregation [28].

Lastly, biofouling occurs when microorganisms, such as bacteria, fungi, and algae, attach to and grow on the membrane surface or within its pores [19]. It is typically caused by the presence of organic matter in the feed stream and by the formation of a suitable environment for microbial growth. Microorganisms can form biofilms, which can further trap other microorganisms and organic and inorganic particles, reducing the membrane’s performance and increasing its pressure drop [19]. For example, extracellular polymeric substances (EPSs) play a critical role in membrane bioreactor (MBR) fouling behavior [29]. EPSs are complex organic compounds produced by microorganisms, primarily in the form of biofilms, and they are usually present in the mixed liquor-suspended solids within MBRs. They have a high molecular weight and can form a gel-like layer on the membrane surface. This gel layer acts as a physical barrier, reducing the permeability of the membrane and causing flux decline. Moreover, EPSs contain hydrophilic and hydrophobic regions, allowing them to interact with both water and solid surfaces. These substances can adhere to the membrane surface due to electrostatic interactions, hydrogen bonding, and van der Waals forces [29,30]. Additionally, EPSs exhibit cohesion forces among themselves, leading to the formation of larger foulant structures. Various strategies can be employed to mitigate the fouling caused by EPSs, such as controlling microbial activity through proper operation and maintenance, optimizing operational parameters (e.g., aeration and hydraulic conditions), employing appropriate pretreatment methods to remove or minimize EPSs, and implementing membrane cleaning protocols. Soluble microbial products (SMPs) can also contribute to fouling in an MBR [31]. SMPs are organic compounds, primarily of microbial origin, that are released into the liquid phase during the metabolic activities of microorganisms. They can accumulate on the membrane surface and form a foulant layer, consisting of proteins, polysaccharides, nucleic acids, and other organic compounds [32]. Thus, understanding the fouling behaviors of EPSs and SMPs in MBRs is essential for developing strategies to mitigate fouling, improve membrane performance, and ensure the reliable operation of MBR systems.

The implications of membrane fouling include a decrease in the membrane flux and an increase in the required transmembrane pressure, which lead to higher energy consumption, a shorter membrane lifespan, and reduced product quality [21]. Therefore, controlling fouling in membrane-based separation processes is essential to ensure optimal performance, a prolonged membrane lifespan, and consistent product quality [33,34]. Furthermore, monitoring the extent and nature of fouling is essential for determining the appropriate cleaning and maintenance protocols. Parameters such as transmembrane pressure, flux decline, and foulant concentration can be monitored to detect fouling and assess the effectiveness of fouling control strategies [35]. Some strategies for controlling and mitigating membrane fouling include the following:Water pretreatment is an essential step to remove suspended solids, organic matter, and other foulants that can clog or damage the membrane surface. Pretreatment methods include sedimentation, coagulation/flocculation, microfiltration/ultrafiltration, and dissolved air flotation [36].The operating conditions of the membrane-based separation process can have a significant impact on fouling. Factors such as the feed flow rate, cross-flow velocity, pressure, pH, and temperature can be optimized to reduce the extent of fouling [35].Antiscalants and dispersants can be added to the feed stream to inhibit the precipitation and accumulation of foulants on the membrane surface. These chemicals work by modifying the surface chemistry of the membrane or by sequestering the foulants in the bulk solution [35].The surface chemistry of the membrane can be modified to reduce fouling by introducing hydrophilic or charged functional groups that repel foulants or by creating surface structures that promote shear-induced turbulence [35].Regular membrane surface cleaning is necessary to remove accumulated foulants and restore the membrane performance. Cleaning methods include physical cleaning, such as backwashing and air scouring, and chemical cleaning, such as acid cleaning and enzyme cleaning [37,38].

## 3. Membrane Fouling Prediction Models

### 3.1. Conventional Models

It is important to predict fouling early in membrane-based separation processes, as this can help prevent or mitigate fouling before it becomes severe and irreversible. The early detection of fouling can help maintain optimal membrane performance, sustain the membrane performance, and reduce the cost of maintenance and replacement [13]. If fouling is left untreated or detected too late, it can lead to irreversible damage to the membrane surface, which may incur hefty costs [11]. As mentioned earlier, timely fouling prediction can be achieved by monitoring key parameters, such as transmembrane pressure, flux decline, and foulant concentration [11]. Predictive models for anticipating the development of fouling can also help predict fouling based on feed stream characteristics, operating conditions, and membrane properties [18]. Table 1 summarizes the advantages and disadvantages of currently available fouling prediction models [20]. While this review focuses on ANN-based models, we will briefly mention the other traditional models herein.

Empirical models are statistical models developed based on experimental data, without necessarily considering the fouling process’s underlying physical or chemical mechanisms [14]. Empirical models for membrane fouling prediction are based on the relationship between the fouling rate and one or more operational parameters. These models are typically simpler to develop and require fewer computational resources than mechanistic models [20]. For example, Darcy’s law model assumes that the resistance to flow across the membrane is proportional to the thickness of the fouling layer [18]. The model predicts the change in flux over time as a function of the filtration time and the thickness of the fouling layer. Another famous model is the Hermia semi-empirical model, which considers the accumulation of particles on the membrane surface and their interaction with the pore structure. The model predicts the fouling resistance as a function of the operating conditions, such as the cross-flow velocity and feed concentration [18].

On the other hand, mechanistic models for predicting membrane fouling are based on the fundamental physical and chemical mechanisms that govern the fouling process [39]. These models consider the mechanisms of fouling, such as deposition, pore constriction, and cake filtration, as well as the effects of operating parameters, such as transmembrane pressure, feed flow rate, and feed concentration [20]. While these models are more complex than empirical models, they offer a more detailed understanding of the fouling process and can be used to design more efficient membrane filtration systems. However, they require a detailed understanding of the underlying physical and chemical mechanisms of fouling, and they may not apply to all membrane filtration systems. Some examples of mechanistic models for membrane fouling prediction include the colloidal fouling model, which considers the interactions between colloidal particles and the membrane surface [26]. The model expresses the fouling rate as a function of the properties of the colloidal particles, such as size, charge, and concentration, as well as the properties of the membrane surface, such as surface charge and hydrophobicity [26]. Another model is the resistance-in-series model, which is based on the concept of hydraulic resistance in series, where the total resistance to filtration is the sum of individual resistances due to fouling mechanisms such as deposition and pore constriction [40]. The model expresses the relationship between the transmembrane pressure and the fouling resistance as a function of the operating parameters. Details of such theoretical models can be found in Yang et al.’s review [20].

Conventional models, such as the resistance-in-series model or the cake filtration model, are often based on well-established principles and mechanisms and can provide more interpretability; they rely on simplified assumptions that may not accurately capture the dynamics of membrane fouling in real-world applications [41]. Additionally, conventional models typically require knowledge of a priori information about the system, such as the nature of the foulants or the fouling mechanism, which may not always be available [41]. Therefore, AI-based models have been increasingly used to predict membrane fouling because they can learn from data, find complex patterns, and make accurate predictions [42,43]. They can be trained on historical data of membrane filtration processes, including operating conditions, membrane properties, and feed characteristics, to predict the fouling rate [17]. While the interplay and competition effects among parameters that affect fouling may not be directly revealed by the ANNs themselves, the overall performance of the antifouling membrane under different conditions can be assessed through their predictive capabilities. The interplay and potential competition effects can be indirectly inferred by systematically varying the operating parameters (e.g., transmembrane pressure, flow rate, and feedwater characteristics) in the data and observing the corresponding changes in the predicted fouling behavior. Sensitivity analysis techniques can be employed to quantify individual parameters’ influence on the ANN model’s output predictions [44]. The factors that have the most significant impact on fouling behavior can be identified by measuring the model’s sensitivity to specific parameter changes. This information helps understand the operating parameters’ relative importance and potential interactions. Thus, combining the predictive power of ANNs with other analytical techniques and domain expertise can lead to a more holistic understanding of the complex dynamics involved in membrane fouling. An upsurge in interest in ANNs for membrane fouling prediction, proven by the increase in the number of publications over the years (Figure 1), can be attributed to several factors, like the ability to develop more sophisticated ANN architectures, improved training algorithms, and increased computing power, which have greatly enhanced the capabilities of ANNs in modeling complex systems. These advancements have spurred interest in exploring the use of ANNs for membrane fouling prediction, as they offer the potential to capture intricate fouling patterns and improve prediction accuracy. Moreover, the accumulation of fouling datasets over the years, through both laboratory-scale experiments and field studies, has facilitated the development and validation of ANN models for membrane fouling prediction. These datasets provide researchers with the necessary information to train and evaluate ANN models, thereby supporting the increasing interest in this research area. Likewise, the complexity of membrane fouling necessitates interdisciplinary collaboration between researchers in fields such as membrane science, water engineering, data science, and AI. This collaboration has fostered the exchange of knowledge and expertise, leading to the integration of ANN techniques into the domain of membrane fouling prediction [17]. Table 1 provides a comparison among the three prevailing types of fouling modeling.

A study by Liu and Kim [46] aimed to compare mathematical blocking laws with an AI-based model for predicting the fouling mechanism in a synthetic water filtration system. The model inputs included the run time, feed turbidity, inlet permeate flow rate, and outlet transmembrane pressure. While insights into membrane fouling mechanisms can be obtained from the blocking laws, a single blocking law cannot adequately fit the entire experimental period. The results of this study indicated that the AI model was more efficient than blocking laws for predicting/simulating complex membrane fouling processes. This and other examples show that AI-based models can incorporate empirical knowledge about the process without explicitly formulating its physical relationship [46]. Another comparative study by Khayet et al. [47] compared the performance of response surface methodology (RSM) versus an AI-based model in predicting RO desalination performance under various operating conditions. When comparing the models’ permeate flux predictions with experimental data, the AI-based model exhibited higher accuracy than the RSM-based model, with a coefficient of determination (R^2^) exceeding 0.9998, indicating excellent experimental data predictability.

**Table 1 membranes-13-00685-t001:** Advantages, disadvantages, and examples of currently available fouling prediction models.

Model Type	Advantages	Limitations	Examples	Ref.
Empirical models	- Useful for predicting fouling in a specific process under certain conditions - Can be simple or complex, depending on the number of parameters included - Relatively easy to develop - Can be useful for process optimization	- Inability to generalize to other processes - Extensive dependence on the quality of experimental data	Darcy’s law model and the fouling index model	[41,48]
Mechanistic models	More accurate and applicable to a wider range of processes	- More complex to develop - require extensive experimental data for calibration	Cake filtration model and pore-blocking model	[39,41,48]
Artificial intelligence models	- Can capture nonlinear relationships between process variables and fouling - Useful for predicting fouling in complex systems	- Require a large amount of training data - May lack interpretability, making them less suitable for process optimization	Support vector machines (SVMs)	[41,49,50]

### 3.2. AI-Based Models

As shown in Figure 2, there are two broad categories of AI models that differ in how they are trained and the types of problems they can handle [51]: supervised and unsupervised. Supervised learning is a type of machine learning in which the model is trained using labeled data, where each data point is associated with a specific output or target variable. Supervised learning aims to learn the relationship between the input and output variables so that the model can accurately predict the output variable for new, unseen data [52]. Supervised learning algorithms include linear regression, logistic regression, decision trees, random forests, and artificial neural networks [52]. On the other hand, unsupervised learning is a type of machine learning in which the model is trained using unlabeled data, where the target variable is unknown. Unsupervised learning aims to identify patterns, structures, or relationships in the data that are not immediately apparent, such as clustering or dimensionality reduction [51]. Unsupervised learning algorithms include k-means clustering, hierarchical clustering, principal component analysis (PCA), and autoencoders [51]. Furthermore, regression, classification, clustering, and dimension reduction are all common types of machine learning problems that can be solved using various AI models [51]:Regression is a type of supervised learning problem in which the goal is to predict a continuous output variable. The model learns the relationship between input variables and output variables using labeled training data and then makes predictions on new, unseen data. Linear, polynomial, and support vector regression are common types of regression algorithms.Classification is another type of supervised learning problem that aims to predict a categorical output variable. The model learns the relationship between input and output variables using labeled training data and then assigns new data points to specific categories based on the learned rules. Common classification algorithms include logistic regression, decision trees, random forests, and support vector machines.Clustering is an unsupervised learning problem that aims to group similar data points into clusters. The model does not use labeled data, but instead finds patterns and structures in the data to group similar data points together. K-means clustering and hierarchical clustering are common clustering algorithms.Dimension reduction is a technique used to reduce the number of input variables in a dataset while still retaining important information. The goal is to simplify the data and remove noise or redundant features that may hinder learning. Principal component analysis (PCA) and autoencoders are common dimension reduction algorithms. While most dimension reduction problems are unsupervised, they can be supervised depending on the specific problem and approach.

The choice of whether to use a supervised or unsupervised AI model for membrane fouling prediction depends on the specific problem and available data [51]. A supervised model may be appropriate if a labeled dataset with known input–output pairs is available. If there is no labeled dataset and/or the goal is to identify patterns and trends in the data, an unsupervised model may be more appropriate [53]. For example, support vector machines (SVMs) are supervised learning algorithms that separate the input data into different classes by finding the hyperplane that maximally separates the classes. SVMs have been applied to membrane fouling prediction by mapping the input features, such as feed concentration, pH, temperature, and membrane properties, to a higher-dimensional space and predicting the fouling rate based on the location of the input data in the feature space. These models are particularly useful for small datasets and can handle noise and outliers in the data [53,54]. However, they may not perform well when the data are highly nonlinear or imbalanced. Another supervised algorithm is decision trees, which construct a tree-like model of decisions and their consequences based on the input data. They have been used for membrane fouling prediction by constructing a tree model of the fouling mechanisms based on the input features, such as the shear rate, membrane pore size, and fouling layer properties. Decision trees are easy to interpret and visualize and can handle missing values and categorical data. However, such models may suffer from overfitting and bias toward the dominant features in the data [55].

Although AI-based models are flexible, adaptable, and accurate, which make them attractive tools for optimizing the performance of membrane-based water treatment systems, they also have limitations that must be considered. For example, AI-based models can be more complex than conventional models, requiring more advanced techniques and expertise to develop and interpret. Moreover, these models require large amounts of high-quality data for training, which may not always be available, especially for the filtration of new or emerging contaminants [56,57]. Consequently, AI-based models can be computationally intensive, requiring significant computational resources, time, and expertise to develop and train. Moreover, these models can be considered “black box” models, meaning that the underlying mechanisms that govern their predictions are not always clear. Additionally, AI-based models may be susceptible to overfitting, where the model becomes too complex and fits the training data too closely, resulting in poor generalization of new data. Lastly, AI-based models heavily rely on the quality and representativeness of the data used for training. If the training data do not represent the real-world system, the model may produce inaccurate or biased predictions [56,57]. While this general topic is extensive, this review will focus on the subset of supervised artificial neural networks (ANNs), a type of supervised machine learning algorithm, that can be used for membrane fouling prediction by training on a labeled dataset with known input–output pairs to learn the relationship between input parameters and the fouling rate [56,58].

## 4. Basic Concepts of ANNs

Several research studies [59,60,61] have utilized smart models to predict membrane fouling indices such as transmembrane pressure (TMP) or membrane permeate flux. These models are based on AI and aim to provide higher accuracy than mechanistic models while at the same time eliminating the need for model calibration. ANNs have been widely used for fouling prediction in membrane filtration systems. Input parameters include feed water quality, operating conditions, and membrane properties. ANNs have several advantages that make them a popular and effective choice for fouling prediction, including [62,63]: (1) the nonlinear modeling capability to capture nonlinear relationships between input and output variables [64], (2) the ability to effectively handle high-dimensional data involving a large number of input parameters, (3) generalizability, as ANNs can be trained on a dataset and then used to predict fouling for new input parameters that were not present in the training set, making them useful for real-time fouling prediction in industrial applications, and (4) flexibility, as they can be easily adapted to different fouling prediction tasks by adjusting the network architecture and input parameters [65].

The basic structure of an ANN consists of a set of artificial neurons linked together to exchange signals. The neurons are sequentially connected as input, hidden, and output layers. Input layers receive input data and pass it on to the hidden layers, which process the data through a series of nonlinear transformations, allowing the network to learn complex patterns in the data [66]. It is an iterative process of trial and error to decide the optimal number of hidden layers, as increasing the number of hidden layers usually increases accuracy until a certain limit, after which overfitting occurs. Barello et al. [67] showed that a time-based ANN model with 2, 3, or 5 hidden layers can produce predictions that are in agreement with correlation data, while 10 and 20 layers result in inaccurate predictions. The net input signal can be computed by summing the input signals multiplied by a random factor called the connection weight, adding to the bias value. Training algorithms, such as Levenberg–Marquardt and gradient descent (GD), tune the weights and biases to reduce errors and boost the accuracy of simulations [58,68]. Then, the output layers produce a predicted output based on the processed input data, which can be computed by multiplying the net input by an activation function, according to Equation (1) [69].
(1)$$Y=f\left(\sum_{k=0}^{n}x_{k}w_{k}+b\right)$$
where *Y* is the output signal, *f* is an activation function, $x_{k}$ is the input data, $w_{k}$ is the connection weight, and *b* is a bias.

The activation function, also known as the transfer function, transfers the signals from the input to the output domain. Two of the most commonly used transfer functions for fouling prediction applications are the linear and sigmoidal functions, shown in Equations (2) and (3), respectively [69].
(2)$$F\left(x\right)=x$$
(3)$$F\left(x\right)=\frac{1}{\left(1+e^{-x}\right)}$$

Several types of ANN architectures have been used for fouling prediction, including feed-forward neural networks (FFNNs) such as radial basis functions (RBFs) [70,71,72], multilayer perception (MLP) [72,73], and recurrent networks (RNNs) [74,75,76]. A comprehensive comparison between MLP and RBF networks was studied by Xie et al. [77]. The choice of architecture depends on the specific problem and available data [66,78]. In the work of Soleimani et al. [79], an FFNN was utilized to predict membrane fouling in oily wastewater. The optimized process parameters included temperature, transmembrane pressure, pH, and velocity. The model’s predicted outcomes showed high accuracy, with an *R*^2^ value greater than 0.99, when compared to the experimental and trained data. Similarly, Rahmanian et al. [80] suggested that using an FFNN model can successfully decipher nonlinear relationships among datasets and accurately predict fouling in UF processes. These studies underscore the suitability of FFNN models for predicting fouling behavior in different contexts. The FFNN architecture allows for modeling complex, nonlinear relationships between input variables and fouling outcomes. The high accuracy achieved in both studies suggests the potential of FFNN models as valuable tools for understanding and mitigating membrane fouling in various applications.

Several available codes and software packages can be utilized to conduct fouling prediction in membrane systems. According to a comprehensive literature survey, MATLAB is the most widely used programming language and software platform, as it offers various toolboxes and functions for data analysis and modeling. It provides flexibility in implementing and customizing ANNs for fouling prediction models. However, MATLAB is a proprietary software, requiring a license for full functionality. Several packages and toolboxes can be useful for membrane fouling prediction using ANNs in MATLAB, as described in Table 2. Utilizing these packages allows the built-in functions and tools to implement, train, and evaluate ANNs for membrane fouling prediction, facilitating the development of accurate and reliable models [81,82]. Another option is Python, an open-source programming language that has gained popularity in scientific computing and data analysis. It offers numerous libraries and packages, such as NumPy, SciPy, and scikit-learn, which provide extensive functionalities for machine learning and predictive modeling [81,82]. When selecting a code or software package for fouling prediction, it is important to consider factors such as availability, cost, flexibility, ease of use, support, and compatibility with existing data and models. Each option has advantages and disadvantages, and the choice depends on specific requirements, expertise, and resources available to the user.

ANNs are developed through a process known as training, which involves adjusting the weights and biases of the network to minimize the difference between the predicted output and the actual output for a set of training examples [86]. The steps are summarized in Figure 3. The first step in developing an ANN is to define the problem that the network will be used to solve. This involves specifying the input variables (e.g., operating conditions, feed characteristics, and membrane characteristics) and the output variable (e.g., fouling rate, and flux decline). Once the problem is defined, data must be collected and preprocessed. This involves cleaning the data to remove any inconsistent or outlier data and then splitting the data into a training set (used to train the network) and a testing set (used to evaluate the network’s performance). Next, the network architecture must be selected based on the problem and the available data. This involves deciding the number of layers and nodes in each layer, as well as the activation functions to be used. The weights and biases of the network are initialized randomly before training begins. During training, the network is presented with a set of input–output pairs from the training set, and the weights and biases are adjusted using an optimization algorithm (e.g., backpropagation) to minimize the difference between the predicted output and the actual output. After training is complete, the performance of the network is evaluated using the testing set. If the performance is satisfactory, the network can be used for predictions on new data. Finally, the network can be fine-tuned and optimized by adjusting the network architecture, training parameters, and preprocessing methods to improve performance [68,86].

In an ANN model for membrane fouling prediction, the input parameters are the variables that are used to predict fouling, while the output parameter is a determinant of fouling propensity, such as the permeate flux decline, fouling growth rate, fouling resistance, or interface energy in MBRs (Figure 4). Interfacial energy plays a crucial role in determining the degree of membrane fouling predicted by ANNs in MBRs. The short-range interfacial force or energy between the foulant and the membrane plays a crucial role in determining the extent of foulant adhesion to the membrane surface. Consequently, accurately quantifying this short-range force or energy holds immense significance in the realm of membrane fouling control [87]. The choice of predicted output parameter depends on the specific application and output availability. Still, the goal is to predict the fouling behavior of the membrane to optimize the membrane performance and extend its lifetime. The selection of input parameters depends on the specific problem and the available data but typically includes operating conditions (e.g., cross-flow velocity, temperature, transmembrane pressure, filtration time, aeration) and/or factors that influence fouling, such as feed characteristics (e.g., pH, ionic strength, particle size distribution, microbial community, organic matter content) or membrane properties (e.g., pore size, material, configuration, hydrophobicity, surface charge) [51]. The material properties of the membrane, such as mechanical strength, surface characteristics, and chemical composition, can significantly impact its fouling behavior and overall performance. These properties influence factors like adhesion, surface interactions, and resistance to fouling agents [88]. Therefore, a comprehensive understanding of the material properties is necessary for accurate predictions and reliable assessments of membrane performance. Without considering the membrane’s material properties, evaluating the long-term durability, stability, and effectiveness of the membrane in real-world applications becomes challenging. Likewise, the number and selection of input parameters can significantly impact the accuracy and performance of the ANN model. Generally, a larger number of input parameters can lead to better predictions but can also increase the complexity of the model and make it more difficult to train. Thus, careful consideration must be given to selecting the most relevant and informative input parameters or predicting fouling in each application [89].

The accuracy of an ANN model for membrane fouling prediction can be evaluated using various statistical metrics, depending on the specific problem and the desired output. The most commonly used assessment metrics include the mean squared error (*MSE*), root mean squared error (*RMSE*) (Equation (4)), mean absolute error (*MAE*), and the coefficient of determination (*R*^2^) (Equation (5)) [51,73,90]. The *RMSE* measures the average of the squared differences between the predicted and actual values of the output parameter; the *MAE* shows the average absolute difference between the predicted and actual values of the output. Lower values of *MSE*, *RMSE*, and *MAE* indicate better model performance. Similarly, *R*^2^ measures the proportion of variation in the output parameter explained by the model, where higher values indicate better model performance. Accuracy, sensitivity, and specificity are also commonly used for classification problems and, respectively, measure the percentage of correctly classified instances, true positive instances, and true negative instances [91].
(4)$$RMSE=\sqrt{\left(\frac{1}{n}\right)}\;\sum_{k=0}^{n}{\left(y_{p}^{\left(k\right)}-y_{k}^{*}\right)}^{2}$$
(5)$$R^{2}=1-\left(\frac{\sum_{k=0}^{n}\left(y_{k}^{*}-y_{p}^{\left(k\right)}\right)}{\sum_{k=0}^{n}\left(y_{k}^{*}-\bar{y}\right)}\right)$$
where $y_{k}^{*}$ is the *k*th target response, $y_{p}^{\left(k\right)}$ is the *k*th predicted response, and $\bar{y}$ is the average *y* value over the range of *n* data points [51].

## 5. Applications of ANNs for Membrane Fouling Prediction

### 5.1. Membrane Fouling Prediction in RO and NF Processes

Roehl et al. [92] developed an MLP ANN model to predict RO membrane fouling. To train the model, the researchers used historical big data consisting of a 76-month process database from a large-scale RO system employed in a wastewater treatment plant, in which they entered 59 hydraulic and water quality parameters. The model was used to calculate the fouling factor, best fouling predictors, and optimization of feed contents to predict fouling growth. According to the model, the best fouling predictors for early-stage fouling included total chlorine content, electrical conductance, and total dissolved solids content; however, late-stage fouling was best predicted by the turbidity, nitrate, nitrite, and organic matter contents of the feed. The outcomes of the models showed high agreement with the experimental data. A critical element of this study is claimed to be the use of a comprehensive experimental approach, resulting in generalized solutions that are expected to have universal relevance to RO-based wastewater treatment. By identifying specific fouling predictors for different fouling stages, the model provides valuable insights into the underlying fouling mechanisms and aids in optimizing operational parameters to mitigate fouling.

The findings of this study have implications for improving the efficiency and performance of RO systems in wastewater treatment, ultimately leading to more effective water purification processes. Moving forward, it would be beneficial to validate the model’s performance across a wider range of wastewater treatment plants and explore its potential application in different water treatment contexts, further establishing its universal relevance in the field. Additionally, considering the dynamic nature of fouling, future research could explore the integration of time-dependent variables and incorporate real-time monitoring data to enhance the accuracy and responsiveness of the ANN model for predicting RO membrane fouling.

Shim et al. [93] developed a long short-term memory (LSTM) model representative of an RNN model to study the filtration performance and predict fouling growth in an NF system. The input parameters were operation time, pressure, initial permeate flux, dissolved organic carbon content, and OCT images of the fouling layer thickness to predict fouling layer thickness growth and permeate flux. The model exhibited a high prediction accuracy for the flux (*R*^2^ = 0.9982) and the fouling layer thickness (*R*^2^ = 0.9987).

Shetty and Chellam [94] tested the accuracy of an ANN model trained using the Levenberg–Marquardt algorithm in predicting fouling in the municipal water NF process. The model inputs—feed pH, ultraviolet absorbance, and total dissolved solids—were used to predict long-term total resistance to water permeation in flat membrane sheets, single spiral-wound elements, and large-scale multiple-stage systems. The model’s predictions were compared against experimental data collected from 11 municipalities using six different membrane types. Collectively, 4 years’ worth of data was used to train, test and validate the model in this study. The researchers aimed to minimize the dataset needed for training yet provide enough data to reach the maximum performance of the model in the testing phase. The prediction accuracy was evaluated based on the *RMSE*, absolute relative error, paired *t*-tests, and Wilcoxon rank sum statistical analyses. For example, they used 10% of the dataset to train the model and obtained a 93% prediction accuracy when tested on a single spiral-wound NF process. Remarkably, they also successfully trained a model using only 4.4% of the dataset. They obtained a 100% prediction accuracy when tested on fouling data from a full-scale process at Palm Beach County, FL. Despite variations in feed water quality and foulant characteristics across different seasons and locations, the ANN successfully predicted experimental observations with less than 5% absolute error in all cases. Upon comparison of the experimental results with the ANN model calculations, they found that the ANN model is a valuable tool for forecasting the membrane fouling behavior. These studies contribute to advancing membrane fouling prediction and highlight the potential of ANN models in optimizing membrane systems for various applications. Future expectations include the further refinement of ANN models and the integration of other analytical techniques to enhance predictive capabilities and broaden the applicability of these models in real-world scenarios.

Park et al. [95] developed a deep neural network (DNN) to model membrane fouling in hybrid NF/RO filtration processes by utilizing optical coherence tomography (OCT) in situ fouling image data and convolutional neural networks (CNNs). The performance of the image-based fouling prediction DNN model was compared with existing mathematical models: the constant resistance model (CRM) and the pore-blocking model (PBM). In total, 13,708 high-resolution fouling layer images were used to develop the DNN model and validate the model performance. The DNN model was trained to simulate both organic fouling growth and flux decline, and it reproduced two- or three-dimensional images. The model showed better predictive performance than the existing mathematical models, as it achieved an *R*^2^ value of 0.99 and *RMSE* of 2.82 μm for the fouling growth simulation and an *R*^2^ of 0.99 and *RMSE* of 0.30 Lm^−2^ h^−1^ for the flux decline simulation. This was a good verification experiment demonstrating that the data-driven approach is an alternative way to model membrane fouling and flux decline processes.

This study also highlighted the potential of data-driven approaches as an alternative to traditional mathematical models, demonstrating the ability of DNNs to automatically learn hierarchical representations from data and uncover hidden features and relationships in complex fouling mechanisms. By utilizing multiple layers of interconnected nodes, DNNs can learn increasingly abstract and high-level data representations, enabling them to accurately capture intricate fouling mechanisms and predict fouling behavior that may not be easily discernible through traditional modeling approaches. Accordingly, the performance of DNNs heavily relies on the availability of high-quality and representative training datasets. Adequate data preprocessing, including feature engineering and normalization, is crucial for optimizing the model performance. Additionally, the architecture and hyperparameter tuning of DNNs requires careful optimization to prevent overfitting or underfitting and to ensure model generalization. The contributions of this study lie in advancing our understanding of membrane fouling processes and providing a framework for utilizing DNNs to accurately predict fouling behavior, ultimately enhancing the design and operation of hybrid NF/RO filtration systems.

Table 3 summarizes some studies that used ANNs to predict fouling in RO and NF systems.

### 5.2. Membrane Fouling Prediction in MF and UF Processes

Liu et al. [60] assessed how well ANN models perform under operating conditions that resemble those used in MF plants. To achieve this, the researchers used hollow-fiber membrane modules in constant flux mode, with periodic backwashing, and simulated the characteristics of surface water using synthetic water. The experimental results and operating parameters were gathered and used as a training dataset for the ANN model. The model architecture consisted of one input layer, two hidden layers, and one output layer, where the input parameters were permeate flux, feed turbidity, operation time, feed ultraviolet absorbance, backwash frequency, and whether the backwash mode was enhanced or normal. It was concluded that the feed quality and operational parameters are of equal importance in predicting the transmembrane pressure profile.

Similarly, Hwang et al. [69] used an MLP model with a backpropagation training algorithm to predict fouling rates in a pilot-scale MF system for filtering water from the Han River. The input parameters included water quality parameters such as turbidity, pH, temperature, total organic carbon, and total algae content, as well as operating conditions like run time and coagulant dosing. The agreement of the model’s outcomes with the experimental data was determined based on the *RMSE* and *R*^2^ values. The model accurately predicted the fouling rate and permeability, with *R*^2^ values of 0.92 and 0.94, respectively.

Future expectations from the studies by Liu et al. [60] and Hwang et al. [69] include further advancements and applications of ANN models for predicting fouling in membrane filtration systems. Both studies highlight the significance of input parameters related to water quality and operational conditions in accurately predicting fouling rates and permeability. Moving forward, there is a potential for enhancing the models by incorporating additional variables or refining the model architectures to improve prediction accuracy.

In another study [110], the effect of polydisperse suspensions on the fouling propensity of MF membranes was investigated. The feed concentration, inlet permeate flux, feed shear rate, run time, and transmembrane pressure were fed as inputs to a basic ANN model consisting of one hidden layer. Statistical analysis at a 95% confidence level using paired *t*-tests and Wilcoxon rank sum tests showed no significant disparities between the outcomes obtained from the ANN model and the experimental data. The authors concluded that the model adequately depicted how intricate particle transportation and deposition mechanisms, along with changes in cake morphology and permeability as a function of variations in the shear rate, permeate flux, and particle size distribution (PSD), can collectively influence membrane fouling. The future directions of this work could involve expanding the scope of the study to explore the impact of additional factors on membrane fouling in polydisperse suspensions. Further investigations could include analyzing the influence of specific particle characteristics, such as shape or surface charge, on fouling propensity. Moreover, conducting experiments with a wider range of operating conditions and varying particle size distributions would enhance the model’s robustness and generalize its findings across different scenarios. By refining the ANN model and considering a more comprehensive set of parameters, future research can provide deeper insights into the mechanisms of fouling in polydisperse suspensions and offer valuable guidance for designing and optimizing membrane filtration systems.

Recent work by Ahmed and Mir [111] used a multilayer FFNN to predict the permeate flux in response to variations in the feed’s applied pressure, run time, Cr(VI) concentration, and cetylpyridinium chloride/Cr ratio in an MF process. The model was trained using the Levenberg–Marquardt algorithm and showed *R*^2^ values ranging from 0.97772 to 0.99992 for different inputs, which indicated that the simulated outputs were in close agreement with cake filtration model data and experimental permeate flux data. The results of the model were used to analyze and determine the fouling mechanism, which, along with other information, can be utilized to save time and cost.

Delgrange et al. [112] developed an ANN model to predict UF fouling, where the model was trained using data extrapolated from a UF drinking water pilot plant. Three network structures were developed and tested, where the optimal architecture (i.e., number of hidden layers) was decided based on the least difference between the training and testing phases. Some of the input data contained the feed temperature, turbidity, and permeate flow rate. It was noted that introducing feed resistance before backwashing as an input to the model enabled it to predict resistance in organic-containing water, even without providing any information to the model about the nature of the organic matter. Consequently, the model developed relationships between organic matter content, irreversible fouling, and subsequent pressure drop across the membrane. Moreover, turbidity was the most significant quality parameter in the model input to predict reversible fouling resistance. Future research could expand the model’s applicability to different operating conditions and validate its performance with real-world data from UF systems. The results of this study, along with outcomes from Dornier et al. [113] and Niemi et al. [59], paved the path for future water research ventures into the use of ANNs for UF fouling prediction.

Another study [114] developed a dynamic ANN model to estimate the specific cake resistance as a function of process parameters (i.e., transmembrane pressure, feed turbidity, run time) in a dead-end UF process. The ANN model was used in conjugation with first-principle filtration models to simplify the nonlinear relationships that exist in the variation of the specific cake resistance, constant flux, and feed water characteristics. A model with five hidden layers showed the best performance in accurately predicting specific cake resistance values that matched with experimental data, with the *MSE* value < 0.01. While irreversible fouling was neglected in this work, the model was claimed to be useful for industrial applications if the ANN data collection and training processes were automated. The combined approach allows for a more comprehensive fouling prediction by taking advantage of both empirical knowledge and the ability of the ANN to uncover hidden patterns. The ANN model can serve as a complementary tool to bridge gaps in understanding and improve fouling predictions’ accuracy and robustness. It can help refine and enhance the first-principle model by capturing any underlying patterns or nonlinear relationships that may be missed among the various parameters that impact fouling. Table 4 summarizes some studies that used ANNs to predict fouling in MF and UF systems, respectively.

Niu et al. [17] reported the accuracy of ANNs in terms of *R*^2^ values in predicting membrane fouling in different membrane-based processes from sixty studies (from 2002 to 2021). Their results are reproduced with their kind permission in Figure 5, which provides a comprehensive overview of the accuracy of ANN-based models in predicting various membrane fouling parameters [17]. The most commonly used fouling predictors include the permeate flux and flow rate, total dissolved solids content, rejection, transmembrane pressure, and fouling resistance. ANNs were found to successfully predict RO and membrane distillation (MD) systems’ permeate flux, with *R*^2^ values exceeding 0.97. However, since MBRs are more complicated systems with complex nonlinear influential factors that affect the permeate flux, the *R*^2^ values of ANN predictions ranged from 0.85 to 0.99 [124].

To advance the field, future research efforts should focus on enhancing ANN-based techniques to achieve even more precise predictions of membrane fouling in different membrane-driven processes. This can be accomplished through several avenues. Firstly, expanding the dataset used for training the models by incorporating more diverse operating conditions, membrane types, and feedwater characteristics would improve the models’ robustness and generalizability. Additionally, integrating advanced data preprocessing techniques, such as feature engineering and dimensionality reduction, can help uncover hidden patterns and optimize the input data representation. Furthermore, exploring the potential of incorporating additional process parameters, such as temperature, pH, and specific foulants, into the ANN models could enhance their predictive capabilities and provide a more comprehensive understanding of fouling mechanisms. Finally, considering the dynamic nature of fouling, incorporating time-dependent or dynamic modeling approaches, such as recurrent neural networks (RNNs) or hybrid models combining ANNs with other modeling techniques, may capture the temporal dynamics and improve long-term predictions. Overall, future research should aim to refine and advance ANN-based techniques for membrane fouling prediction to enhance process control, optimize operational parameters, and improve the overall efficiency and reliability of membrane-based processes.

### 5.3. Interfacial Energy Prediction in MBRs

During the operation of an MBR, various foulants can be transported toward the membrane surface due to fluid dynamics induced by agitation, aeration, or filtration. However, the crucial factor determining the ultimate adhesion of these foulants to the membrane surface is the short-range interfacial force or energy between the foulant and the membrane [125,126]. Therefore, accurately quantifying this short-range force or energy holds immense significance in controlling membrane fouling. The extended Derjaguin–Landau–Verwey–Overbeek (XDLVO) theory offers a method to quantify the short-range force or energy between two smooth planes [127]. However, the surfaces of the membranes utilized in MBRs are typically randomly rough. Consequently, the XDLVO theory cannot accurately quantify the short-range force or energy between foulants and the actual membrane surface [128,129]. This highlights the need for alternative methods to assess and quantify the interfacial interactions between foulants and the real membrane surface in MBR systems [130].

It is important to highlight that recent research has incorporated the surface element integration (SEI) method and triangulation technology into the existing XDLVO theory [131]. This integration has led to the development of the advanced XDLVO approach, specifically designed to quantify the interfacial energy between foulants and the actual rough surface of membranes. The integration process required for the advanced XDLVO approach can be achieved through computer programming using platforms like MATLAB [132]. However, it is important to note that this integration process involves ultrahigh computational complexity due to the intricate nonlinear mapping relations between interfacial energy and various factors. Even for simple interaction scenarios, it can take several days to complete the quantification of interfacial energy. As a result, the application value of the advanced XDLVO approach is significantly reduced. Furthermore, the interfacial energy is influenced by various factors, such as the surface properties of the foulants and the membrane, the separation distance, and aqueous solution conditions [133]. Unfortunately, the advanced XDLVO approach cannot accurately quantify interfacial energies in scenarios where these conditions are altered, which hinders its practical application. Nevertheless, ANNs can process complex nonlinear mappings and demonstrate robust capabilities in pattern recognition and data fitting. As a result, ANNs can potentially address the limitations of the advanced XDLVO approach.

Zhao et al. [134] utilized an RBF ANN model to predict the interfacial interactions of sludge foulants with a randomly rough flat-sheet poly(vinylidene fluoride) membrane within a separation distance ranging from 0.158 nm to 10 nm. In this investigation, the interfacial interaction data acquired from the advanced XDLVO approach were split into two sets: a training set and a test set. Once the RBF network was adequately trained, it was employed to forecast interfacial interactions at different separation distances. They concluded that training the network with at least ten samples was sufficient to produce highly trained models that exhibited high calculation accuracy. Moreover, the computation time consumed by the RBF ANN was 1/50 of that consumed by the advanced XDLVO approach to simulate the same conditions.

These findings were also supported by the work of Chen et al., who assessed the viability of utilizing RBF ANN [71], BP ANN, and generalized regression neural network (GRNN) methods [135] for quantifying interfacial energy related to membrane fouling in an MBR. In both studies, the researchers used an input vector consisting of five factors, which were the contact angles of three different model liquids (i.e., ultrapure water, glycerin, diiodomethane) on the foulant surface, the zeta potential of the membrane surface, and the separation distance. Remarkably, the RBF ANN illustrated a high regression coefficient, high accuracy, and a more rapid response than the advanced XDLVO approach when applied to the same dataset [71]. Similarly, both the BP ANN and the GRNN have demonstrated their robustness and ability to effectively capture the relationships between interfacial energy and the various tested factors. However, the BP ANN model prediction performance was superior [135]. BP ANN is an FNN capable of learning and storing numerous input–output mapping relationships without initially revealing the mathematical equations underlying these mappings, which makes it one of the most extensively utilized ANN models. On the other hand, the GRNN possesses nonlinear mapping capabilities, rapid learning speed, and the ability to converge to optimized regression through effective sample aggregation. The GRNN can yield satisfactory prediction results even with limited training samples, making it suitable for handling unstable data. Thus, it is reasonable to expect that both the BP ANN and GRNN would also offer distinct advantages in quantifying interfacial energy [135].

In summary, previous studies on membrane bioreactors have provided valuable insights into interfacial energy prediction and fouling behavior. The advanced XDLVO approach has contributed to understanding the short-range interfacial forces and energy that govern the adhesion of foulants to membrane surfaces. The introduction of ANNs (e.g., RBF, BP, and GRNNs) has shown promise in overcoming the limitations of the advanced XDLVO approach. These ANNs possess the capability to handle complex nonlinear mappings, enabling them to capture the intricate relationships between interfacial energy and various factors associated with fouling. Furthermore, the comparison between the ANN models and the advanced XDLVO approach has demonstrated superior performance in terms of the regression coefficient, accuracy, and response time. These findings suggest that ANNs can be a valuable tool for interfacial energy quantification in membrane fouling studies. Overall, these studies have laid the foundation for further research in developing efficient methods for predicting and controlling membrane fouling in MBRs, ultimately leading to the improved performance and longevity of membrane-based systems in various applications.

## 6. Future Directions and Conclusions

While the application of ANNs for membrane fouling prediction has shown promising results, there are several avenues for further exploration and improvement. Current ANN models for membrane fouling prediction primarily utilize readily available operating parameters, such as transmembrane pressure, flow rate, and feedwater characteristics. Future outlooks should include the following:-Exploring the inclusion of additional input variables that capture other relevant aspects of the system, such as pretreatment methods, membrane characteristics, and fouling mitigation strategies. This expansion of input variables can lead to more comprehensive and reliable prediction models.-Incorporating the real-time monitoring of relevant process variables, such as membrane permeability, fouling resistance, and hydraulic parameters, can provide valuable insights into fouling behavior. Integrating these dynamic data into ANN models can enable the continuous prediction and monitoring of fouling progression. This integration may facilitate proactive fouling management strategies, allowing for timely maintenance actions or the optimization of operational conditions to mitigate fouling.-Integrating analytical techniques to interpret and visualize the learned representations and decision-making processes within ANNs. While ANNs are known for their exceptional predictive capabilities, their black-box nature often limits their interpretability. The ability to explain model predictions and identify key factors influencing fouling can greatly enhance the practical utility of ANN models. This would enable researchers and practitioners to gain valuable insights into fouling mechanisms and optimize operational strategies, accordingly.-Shifting the focus toward their practical deployment and implementation as the ANN models mature and demonstrate their effectiveness in membrane fouling prediction. Researchers can collaborate with membrane manufacturers, operators, and stakeholders to develop user-friendly software tools or decision support systems that incorporate ANN models. These tools can aid in real-time fouling prediction and the early detection of anomalies and support informed decision-making for fouling control strategies.-Incorporating hybrid models to leverage the strengths of multiple modeling techniques allows for a more comprehensive understanding of the complex dynamics involved in membrane fouling. Researchers can harness the complementary advantages of each approach by integrating ANNs with other models, such as empirical, mechanistic, or statistical models. By combining the strengths of different models, hybrid methods can enhance prediction accuracy, generalization capabilities, and the incorporation of domain knowledge. As research in this area continues to evolve, the development and refinement of hybrid models hold significant potential for advancing our understanding of fouling mechanisms and optimizing fouling control strategies in membrane-based processes.

In conclusion, membrane-based processes have emerged as a promising and efficient approach for separation, with a wide range of applications in desalination, water reuse, and wastewater treatment. However, membrane fouling remains a significant challenge in these processes, and further research is needed to develop effective fouling mitigation strategies and optimize the performance of membrane-based processes. ANN and its derivatives have shown great potential in predicting and modeling membrane fouling in various membrane filtration processes. ANNs offer flexibility in handling diverse types of input data, including operating parameters, feedwater characteristics, and membrane properties. This versatility allows for a more holistic understanding of fouling processes and the ability to capture the interactions between various factors that influence fouling behavior. They are also data-driven, meaning they can learn and adapt from the available dataset without being constrained by pre-existing assumptions or theoretical frameworks. In contrast, other theoretical and mathematical models, such as empirical models or mechanistic models, offer a more explicit representation of the underlying physical and chemical processes involved in fouling. These models are often based on well-established principles and equations, allowing for a deeper understanding of the fouling mechanisms and the ability to interpret the results more transparently. By exploring the aforementioned future directions, researchers can advance the field and pave the way for more accurate, robust, and actionable fouling prediction models, ultimately contributing to the optimization and cost-effectiveness of membrane filtration systems.

## Figures and Tables

**Figure 1 membranes-13-00685-f001:**
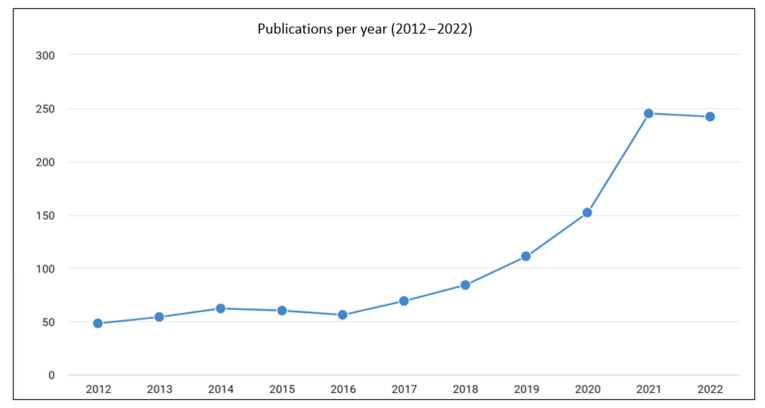
An upsurge in the number of publications involving the use of ANNs for membrane fouling prediction has been observed over the past decade. Figure generated using Dimensions^®^ at https://app.dimensions.ai/discover/publication, accessed on 20 April 2023, with the keywords “ANNs” AND “membrane fouling prediction”, limiting the research to peer-reviewed journal articles [45].

**Figure 2 membranes-13-00685-f002:**
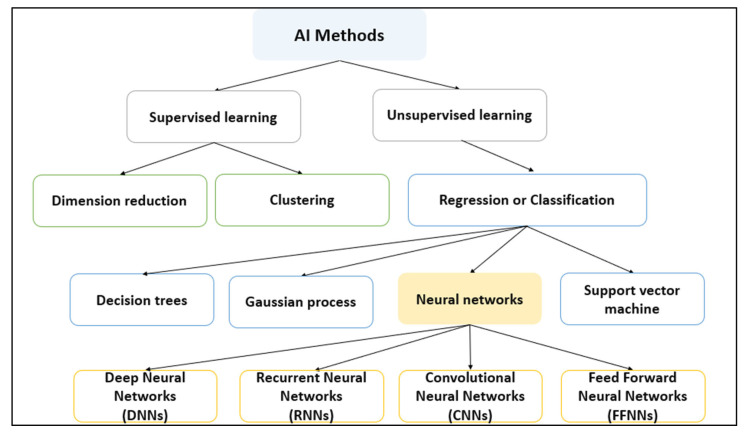
Different AI-based models commonly used for membrane fouling prediction. DNN: Deep neural network; RNN: recurrent neural network; CNN: convolutional neural network; FFNN: feed-forward neural network.

**Figure 3 membranes-13-00685-f003:**
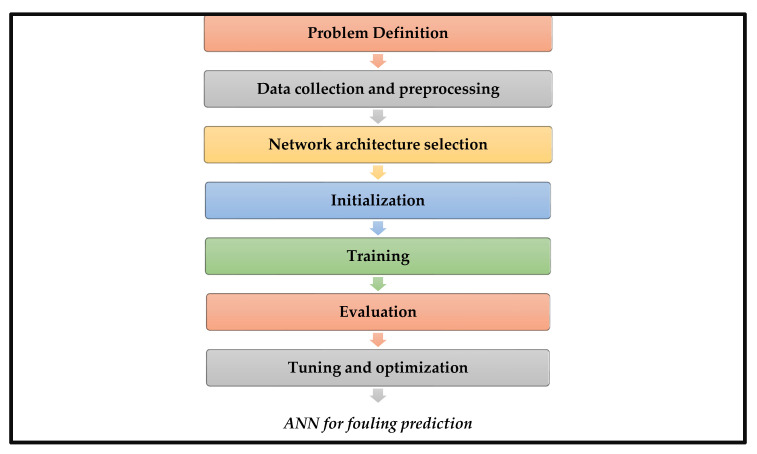
The general process for developing an ANN.

**Figure 4 membranes-13-00685-f004:**
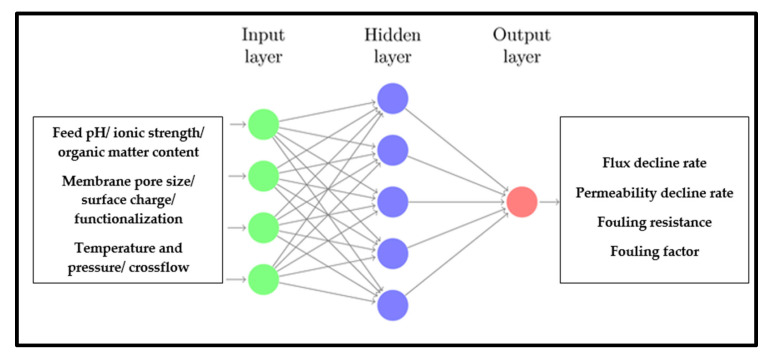
A basic ANN structure with commonly used input parameters and output predictions.

**Figure 5 membranes-13-00685-f005:**
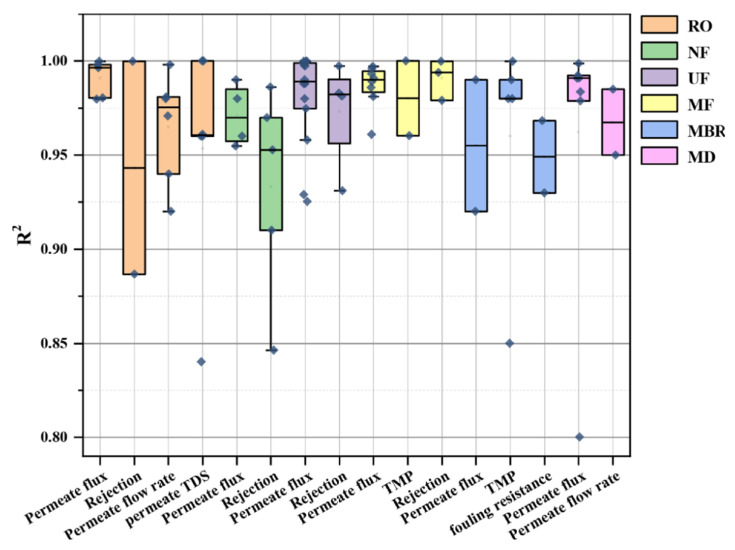
Accuracy of ANNs in predicting membrane fouling in different membrane-based processes. *R*^2^ values from 60 studies (from 2002 to 2021) were used. Reproduced with permission from [17] (Artificial intelligence-incorporated membrane fouling prediction for membrane-based processes in the past 20 years: A critical review), published by Elsevier (2022).

**Table 2 membranes-13-00685-t002:** Available *MATLAB toolboxes* for developing ANNs for membrane fouling prediction [83,84,85].

Toolbox	Advantages	Limitations
Neural Network Toolbox	Provides a comprehensive set of functions and tools specifically designed for neural networks. Supports various types of neural network architectures. Offers advanced training algorithms to improve network performance. Provides visualization tools for network analysis and debugging.	Requires familiarity with neural networks and their implementation. Limited support for other machine learning algorithms outside of neural networks. May require additional customization for specific fouling prediction tasks.
Curve Fitting Toolbox	Offers a wide range of algorithms for curve fitting and regression analysis. Provides functions for optimizing model parameters and assessing model accuracy. Includes tools for model validation and comparison. Supports customization of fitting options for specific fouling prediction models.	Primarily focused on curve fitting tasks and may have limited functionalities for complex ANNs. Requires preprocessed data in a suitable format for curve fitting analysis. May require additional statistical knowledge for optimal model selection and validation.
Optimization Toolbox	Offers a variety of optimization algorithms for parameter tuning in ANNs. Provides tools for fine-tuning network architecture and training parameters. Supports constrained and unconstrained optimization problems. Enables efficient parameter search and optimization of ANN models.	Focused primarily on optimization tasks and may require additional integration with ANN functionalities. May require expertise in optimization algorithms for optimal model configuration. Limited support for handling large-scale optimization problems.
Deep Learning Toolbox	Specifically designed for deep learning tasks, including CNNs and RNNs. Offers pre-trained models and transfer learning capabilities. Provides tools for model visualization, training, and deployment. Includes support for Graphics Processing Unit (GPU) acceleration, enabling faster computations for large datasets.	Primarily focused on deep learning models and may have limited functionalities for other ANN architectures. Requires familiarity with deep learning concepts and techniques. GPU usage may require additional hardware resources.

**Table 3 membranes-13-00685-t003:** Summary of some studies on the use of ANNs fouling prediction in RO and NF processes.

Process Type	ANN Model	Training Algorithm	Feed Type	Model Inputs	Model Outputs	Ref.
RO	Basic ANN	Levenberg–Marquardt	Groundwater	Feed salinity, operating pressure, run time, membrane type	Water permeability constant (*R*^2^ = 0.996)	[67]
	Basic ANN	Bayesian regularization backpropagation	Formulated artificial groundwater	Feed concentration, pressure, temperature	Water recovery (*R*^2^ = 0.9611) Total dissolved solids rejection (*R*^2^ = 0.9246) Specific energy consumption (*R*^2^ = 0.9044)	[96]
	RBF	Orthogonal least squares	Brackish water	Temperature, pH, conductance, pressure	Permeate flow rate (*R*^2^ = 0.9853) Total dissolved solids content (*R*^2^ = 1)	[72]
	RBF	Backpropagation	Groundwater	Clustered input space consisting of an 8-variable vector	Permeate flow rate (*R*^2^ = 0.92) Permeate total dissolved solids content (*R*^2^ = 0.84)	[97]
	MLP	Backpropagation	Wastewater	Run time, total dissolved solids content, feed	Permeate flow rate (*R*^2^ = 0.97–0.99)	[98]
	MLP	Backpropagation	Brackish water	Time, conductivity, flow rate, transmembrane pressure	Permeate flow (*R*^2^ = 0.94) Permeate conductivity (*R*^2^ = 0.99)	[99]
	MLP	Backpropagation	Dilute saline water	Membrane pore radius, friction constants between solvent, solute, and membrane, feed solute concentration, pressure, temperature	Total flux (*R*^2^ = 0.9982) Solvent flux (*R*^2^ = 0.9980) Separation factor (*R*^2^ = 0.9997)	[100]
	MLP	Backpropagation	Brackish water	Feed flow rate, pH, temperature, pressure, feed conductivity	Standardized permeate flux (*R*^2^ = 0.68) Percent salt passage (*R*^2^ = 0.86)	[101]
	MLP	Backpropagation	Seawater	Run time, transmembrane pressure, feed flow rate, feed total dissolved solids content, temperature	Permeate flow rate (*R*^2^ = 0.75) Permeate total dissolved solids content (*R*^2^ = 0.96)	[102]
	MLP	Levenberg–Marquardt	Brackish water	Feed salt concentration, temperature, pressure	Permeate flow rate (*R*^2^ = 0.998)	[103]
	MLP	Levenberg–Marquardt	Brackish water	Temperature, pH, conductance, pressure	Permeate flow rate (*R*^2^ = 0.9904) Permeate total dissolved solids content (*R*^2^ = 1)	[72]
NF	Basic ANN	Backpropagation	Humic acid-based feedwater	Cross-flow velocity, initial flux, feed calcium concentration	Permeate flux (absolute relative error <0.1%)	[104]
	Bootstrap aggregated neural networks (BANNs)	Ensemble creation and aggregation	Organic-contaminated feed	Feed characteristics (dipole moment, molecular weight, zeta potential, pH), recovery %, temperature, pressure	Salt rejection percent (*R*^2^ = 0.9862)	[105]
	DNN	/	Humic acid-based feedwater	Initial fouling thickness, membrane type, time, initial permeate flux	Permeate flux (*R*^2^ = 0.99) Fouling layer thickness (*R*^2^ = 0.99)	[95]
	RNN	Backpropagation	Artificial saline water	Real-time 2-dimensional OCT images, operation time, initial permeate flux, pressure, fluorescence regional integration, feed dissolved organic carbon content	Permeate flux (*R*^2^ = 0.9982) Fouling layer thickness (*R*^2^ = 0.9987)	[93]
	Normalized RBF	Backpropagation	Groundwater	Feed total dissolved solids, feed flux, recovery %, net driving force	Total dissolved solids concentration (*R*^2^ = 0.99)	[106]
	MLP	Backpropagation	Organic solvent feed	Membrane characteristics (support material, molecular weight cutoff), solvent properties (molecular weight, viscosity, density, kinetic diameter, etc.), operating conditions (temperature, pressure, solute concentration, solute type)	Permeance (*R*^2^ = 0.98) Rejection (*R*^2^ = 0.91)	[49]
	MLP	Backpropagation	Highly concentrated salt solutions	Feed pressure, salt concentration	Permeate flux, salt rejection	[107]
	MLP	Backpropagation	Groundwater	Feed total dissolved solids, feed flux, recovery %, net driving force	Total dissolved solids concentration (*R*^2^ = 0.99)	[106]
	MLP	Backpropagation	Organic contaminated feed	Feed characteristics (dipole moment, molecular weight, zeta potential, pH), contact angle, recovery %, temperature, pressure	Salt rejection percent (*R*^2^ = 9527)	[108]
	MLP	Levenberg–Marquardt	Groundwater and surface water	Permeate water flux, feed absorbance, time, pH, total dissolved solids content, temperature, influent flow rate	Membrane fouling resistance (absolute relative error < 5%)	[94]
	MLP	Levenberg–Marquardt	Waste brine	Transmembrane pressure, temperature, run time	Permeate flux (*R*^2^ = 0.96) Fouling resistance (*R*^2^ = 0.98)	[109]

**Table 4 membranes-13-00685-t004:** Summary of some studies on the use of ANNs for fouling prediction in MF and UF processes.

Process Type	ANN Model	Training Algorithm	Feed Type	Model Inputs	Model Outputs (*R*^2^ Values)	Ref.
MF	Basic ANN	Levenberg–Marquardt	Bovine serum albumin solution	Transmembrane pressure, feed pH, cross-flow velocity	Permeate flux (*R*^2^ = 0.9810) Rejection (*R*^2^ = 0.99997)	[115]
	RBF	/	Bovine serum albumin solution	Transmembrane pressure, feed pH, cross-flow velocity	Permeate flux (*R*^2^ = 0.9932) Rejection (*R*^2^ = 0.97903)	[115]
	MLP	Gradient descent with momentum	Red plum juice	Transmembrane pressure, temperature, membrane pore size, feed flow rate, run time	Permeate flux (*R*^2^ = 0.961)	[116]
	MLP	Backpropagation	Particulate suspensions	Influent velocity, feed concentration, transmembrane pressure	Flux improvement efficiency by turbulence promoter (*R*^2^ = 0.9891)	[117]
	MLP	Backpropagation	Wastewater	Temperature, pH, transmembrane pressure, cross-flow velocity, filtration time	Permeate flux (*R*^2^ = 0.9999) Fouling resistance (*R*^2^ = 0.9999)	[79]
	MLP	Backpropagation	Nickel-ion-containing aqueous solution	Membrane material, pore size, adsorbent type, surfactant type, surfactant concentration	Transient flux (*R*^2^ = 0.986)	[118]
	MLP	Levenberg–Marquardt	Industrial oily water	Temperature, transmembrane pressure, cross-flow velocity, feed oil concentration, run time	Permeate flux (*R*^2^ = 0.997)	[119]
UF	MLP	Backpropagation	Wastewater	Surfactant-to-metal ratio, pH, cumulative sampling volume	Permeate flux, rejection rate (*R*^2^ = 0.9974)	[120]
	MLP	/	Organic-contaminated feed	Parallel factor analysis (PARAFAC) component maximum fluorescence, pH, turbidity, and historical average slope of the resistance	Fouling resistance (mean absolute relative error < 5%)	[121]
	MLP	Backpropagation	Oily wastewater (polyacrylonitrile (PAN)-containing feed)	Temperature, pH, transmembrane pressure, cross-flow velocity, filtration time	Permeate flux decline (*R*^2^ = 0.99997)	[122]
	MLP	Levenberg–Marquardt	Bovine serum albumin solution	Membrane pore size, protein solution concentration, pH, transmembrane pressure, cross-flow velocity	Permeate flux (*R*^2^ = 0.996) Rejection (*R*^2^ = 0.994)	[48]
	MLP	Levenberg–Marquardt	Synthetic wastewater containing zinc ions	Surfactant-to-metal ratio, pH, feed anionic surfactant concentration, transmembrane pressure, ligand–to-zinc ratio, electrolyte concentration	Permeate flux (*R*^2^ = 0.929) Rejection (*R*^2^ = 0.981)	[123]
	MLP	Levenberg–Marquardt backpropagation	Polyethylene glycol-containing feed	Transmembrane pressure, cross-flow velocity, operation time	Permeate flux (*R*^2^ = 0.9977)	[50]
	MLP	Bayesian regulation backpropagation algorithm and Levenberg–Marquardt	Wastewater	Temperature, pH, transmembrane pressure, cross-flow velocity, filtration time	Permeate flux (*R*^2^ = 0.9999)	[42]
	MLP	Backpropagation, scaled conjugate gradient, Levenberg–Marquardt, gradient descent with momentum, adaptive learning rate backpropagation	Synthetic wastewater containing lead ions	Surfactant-to-metal ratio, pH, feed anionic surfactant concentration	Permeate flux (*R*^2^ = 0.9254) Rejection (*R*^2^ = 0.9813)	[80]

## Data Availability

No new data were created or analyzed in this study. Data sharing is not applicable to this article.

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
