# Peer review of "A Review on Membrane Fouling Prediction Using Artificial Neural Networks (ANNs)"

_membranes, 2023, doi:10.3390/membranes13070685_

Round 1

Reviewer 1 Report

This reviewed paper provides an overview of membrane fouling prediction models using Artificial Neural Networks (ANNs). The paper highlights the importance of membrane-based processes for water treatment and the challenges posed by fouling. It discusses the limitations of traditional models and the potential of AI-based models, specifically ANNs, for predicting fouling in membrane processes.

However, the paper requires major revisions due to several shortcomings. 

First, the Introduction section would benefit from a graphical representation of the paper's structure and main content. This would enhance the reader's understanding and provide a clear roadmap.

Second, in Section "2. Fouling types and mitigation strategies," important methods and relevant literature are missing. It is recommended to incorporate references such as the Journal of Membrane Science 2014, 460, 110-125; Water Research 2019, 149, 477-487; and Water Research 2021, 189, 116665 to provide a comprehensive review of fouling types and mitigation strategies.

Third, Figure 2 lacks a clear illustration of the structure of an artificial neural network. Adding a schematic representation would enhance the clarity of the presented information.

Fourth, in Section "5. Applications of ANNs for membrane fouling prediction," an important research direction is missing, namely, the interface energy that determines the degree of membrane fouling predicted by artificial neural networks. It is suggested to include a subsection summarizing this direction, with references to Bioresource Technology 2019, 282, 262-268; Journal of Colloid and Interface Science 2023, 640, 110-120; Journal of Cleaner Production 2022, 376, 134236; Journal of Colloid and Interface Science 2020, 565, 1-10; and Bioresource Technology 2019, 282, 262-268.

Fifth, it is recommended to remove some less relevant references that do not contribute significantly to the content.

Finally, in Section "6. Future directions and conclusion," it would be beneficial to present the main directions and conclusions as bullet points for better clarity.

The language should be polished to improve the overall readability and coherence of the paper.

Author Response

Please find the attached rebuttal.

Reviewer 2 Report

Please find the comments in attachment

Author Response

Please find the attached rebuttal letter for reviewer 2.
